# Structural and Functional Neuroimaging of Visual Hallucinations in Lewy Body Disease: A Systematic Literature Review

**DOI:** 10.3390/brainsci7070084

**Published:** 2017-07-15

**Authors:** Stefania Pezzoli, Annachiara Cagnin, Oliver Bandmann, Annalena Venneri

**Affiliations:** 1Department of Neuroscience, University of Sheffield, Sheffield S10 2RX, UK; spezzoli1@sheffield.ac.uk (S.P.); o.bandmann@sheffield.ac.uk (O.B.); 2Department of Neurosciences, University of Padua, 35128 Padua, Italy; annachiara.cagnin@unipd.it; 3Istituto di Ricovero e Cura a Carattere Scientifico (IRCCS) Fondazione Ospedale San Camillo, 30126 Venice, Italy

**Keywords:** visual hallucinations, Lewy body, Parkinson’s disease, Parkinson’s disease dementia, dementia with Lewy bodies, MRI, fMRI, DTI, PET, SPECT

## Abstract

Patients with Lewy body disease (LBD) frequently experience visual hallucinations (VH), well-formed images perceived without the presence of real stimuli. The structural and functional brain mechanisms underlying VH in LBD are still unclear. The present review summarises the current literature on the neural correlates of VH in LBD, namely Parkinson’s disease (PD), and dementia with Lewy bodies (DLB). Following a systematic literature search, 56 neuroimaging studies of VH in PD and DLB were critically reviewed and evaluated for quality assessment. The main structural neuroimaging results on VH in LBD revealed grey matter loss in frontal areas in patients with dementia, and parietal and occipito-temporal regions in PD without dementia. Parietal and temporal hypometabolism was also reported in hallucinating PD patients. Disrupted functional connectivity was detected especially in the default mode network and fronto-parietal regions. However, evidence on structural and functional connectivity is still limited and requires further investigation. The current literature is in line with integrative models of VH suggesting a role of attention and perception deficits in the development of VH. However, despite the close relationship between VH and cognitive impairment, its associations with brain structure and function have been explored only by a limited number of studies.

## 1. Introduction

The term Lewy body disease (LBD) refers to disorders characterised by the neural inclusion of pathologic α-synuclein aggregates called Lewy bodies [1,2]. Clinical manifestations of Lewy body pathology include dementia with Lewy bodies (DLB), Parkinson’s disease (PD), and Parkinson’s disease dementia (PDD). These diseases share some clinical characteristics, including motor symptoms, sleep disorders, cognitive impairment, and visual hallucinations (VH) [3]. VHs represent a common symptom experienced by patients with LBD, and they are among the core features of the DLB symptomatology [4]. Recurrent and complex VHs have been defined as repetitive and well-formed images, which are perceived without the presence of real stimuli [5]. The most commonly reported VHs consist of people, animals, and inanimate objects, which appear to have similar features between clinical conditions, especially DLB and PD with dementia [5,6,7]. Early false sensations of presence are also common, mainly in PD [7]. Patients with LBD may also experience illusions or misperceptions, which are defined as incorrect perceptions of real stimuli [5,8]. VHs seem to be more severe and complex in LBD patients with cognitive impairment, which often present with lack of insight about the unreal nature of their experience [7].

The presence of VHs is a strong predictor of Lewy body pathology at autopsy [9,10]. Lewy body pathology associated with VHs was shown to affect temporal lobe areas predominantly, mainly the amygdala [11,12,13]. The structural and functional brain correlates underlying this symptomatology are, however, still not well understood. Moreover, it is not clear whether different diseases within the LBD spectrum share common neural mechanisms associated with VH, or whether they differ between conditions. Neuroimaging techniques represent valuable tools, which may help in detecting in vivo biomarkers that specifically characterise patients with VHs in PD and DLB. This may help the achievement of a more accurate understanding of the biological vulnerabilities leading to VHs, and may help future prediction of patients who are likely to develop these disabling symptoms, leading to the possible implementation of preventive treatments.

The aim of the present review was to summarise the current literature on structural and functional brain abnormalities associated with VHs in LBD, namely PD and DLB. Specifically, findings from structural magnetic resonance imaging (MRI), diffusion tensor imaging (DTI), functional MRI (fMRI), positron emission tomography (PET), and single photon emission computed tomography (SPECT) studies were critically reviewed.

## 2. Materials and Methods

Articles were identified through a systematic literature search, which was carried out in January 2017 by using the PubMed and Web of Science databases with no time limit. The following key words were used: “visual hallucinations”, “visual hallucination”, “Lewy body", "dementia with Lewy bodies”, “Parkinson’s disease”, “magnetic resonance imaging”, “MRI”, “voxel-based morphometry” (VBM) and “VBM”, “fMRI”, "resting-state", “diffusion tensor imaging”, “DTI”, “positron emission tomography” and “PET”, and “single photon emission computed tomography” and “SPECT”. An additional manual search of references was also undertaken. Studies were excluded according to the following exclusion criteria: (1) pathologies other than DLB, PD, or PDD; (2) neuroimaging analysis not related to VH; (3) patients with medication-induced VH; (4) studies not using MRI, fMRI, DTI, PET, SPECT; (5) PET and studies not investigating glucose metabolism and regional cerebral blood flow; (6) MRI studies using visual rating; (7) magnetic resonance spectroscopic imaging; (8) pharmacological studies; (9) case studies (except for fMRI during VH); (10) review and theoretical articles; (11) non-English articles; and (12) non-peer reviewed articles. The search strategy used followed the PRISMA guidelines [14]. The articles included were assessed for scientific suitability to the aim of the present review, by using a set of 14 criteria adapted from Welton et al. [15], (these criteria are listed in Appendix A). Each article was rated from 0 to 14, assessing the scientific quality of its structural and functional neuroimaging analyses related to VH only.

## 3. Results

The initial search retrieved 646 titles, among which 387 were duplicate publications, which were excluded. Three studies were identified through manual search. A total number of 262 titles and abstracts were assessed, of which 88 full-text articles were retrieved and screened for eligibility. The final review included 56 studies investigating the structural and functional brain correlates of VH in LBD by using structural and functional MRI, DTI, PET, and SPECT. A flow chart describing the selection process of the studies included in the final review is shown in Figure 1 (adapted from Moher et al. [14]).

Studies investigating VH in LBD using more than one approach or imaging technique were included in more than one section of the review and the findings for each technique reported separately in the relevant section. There were eight studies combining different techniques, specifically structural and functional MRI [16,17]; structural MRI and PET [18,19]; structural MRI and DTI [20,21]; structural MRI, DTI, and fMRI [22]; and fMRI and arterial spin labelling (ASL)-MRI [23]. Moreover, in two structural MRI studies, different methods were used to investigate regional brain volumes [24,25].

Suitability assessment of the articles reviewed revealed that structural (mean = 7.92), and functional (mean = 7.06) neuroimaging analyses related to VH were of comparable quality. Overall, among the main limitations of the studies, we found lack of a priori hypotheses on VH (*n* = 27 studies), sample sizes <15 participants per group (*n* = 39 studies), and absence of correlational analyses with VH indices (*n* = 35 studies) and cognitive measures (*n* = 49 studies). Quality assessment of each study can be found in Appendix A.

### 3.1. Structural Brain Imaging

The brain structural changes associated with VH in LBD, detected with MRI, were investigated by 24 studies. The findings are summarised according to the analytic approach used, namely VBM, other methods to investigate brain morphology, namely regional volumes, shape, and cortical thickness; and DTI. For each study, detailed demographic, clinical, and methodological information—and imaging results related to VH—are reported in Appendix B (Table 2).

#### 3.1.1. Voxel-Based Morphometry

A total of 12 VBM studies were identified. Ten studies focused on regional volumetric brain differences between LBD patients with and without VH, including nine studies which used whole brain analyses [21,24,25,26,27,28,29,30,31], and three voxel-based analyses restricted to predefined regions of interest (ROI) [28,32,33]. Four studies also included results on the association between grey matter loss and VH or other cognitive variables [27,32,33,34]. Moreover, one study included only the comparison between PD subgroups and controls [35], and another investigated progression of brain atrophy [34]. The VBM methodology implemented was largely consistent between studies. All used the Statistical Parametric Mapping (SPM) software for imaging analysis, and seven [21,25,26,27,28,31,35] used the Diffeomorphic Anatomical Registration using Exponentiated Lie algebra (DARTEL) algorithm to create a study-specific template. A threshold corrected for multiple comparisons was applied in four whole brain [24,25,30,34], and four ROI [28,32,33,35] studies. The results of those studies which used uncorrected thresholds [21,25,26,27,28,31] should be interpreted more cautiously, since analyses using an uncorrected threshold may generate a higher number of false positives. Clinical and demographic features, including severity of cognitive impairment, disease duration, age, and years of education varied between studies. Some of them reported no differences between patients with and without VH in global cognitive impairment [21,24,25,27,28,31,33], while others showed more severe cognitive decline in hallucinating patients [30,32,34,35]. Furthermore, some studies on PD reported more advanced disease stage (Hoehn and Yahr, H&Y, stage) in patients with VH than in those without [21,28,30,34].

Ibarretxe-Bilbao et al. investigated the progression of brain atrophy in PD patients with and without VH [34]. Hallucinating patients presented more extensive grey matter loss over time, accompanied by faster cognitive decline. Progressive grey matter reduction from baseline to follow-up extended to parietal, temporal, frontal, thalamic, and limbic areas in patients with VH, whereas only small clusters in frontal and cerebellar regions showed reductions in those without. Cognitive impairment and disease severity at baseline, however, were greater in patients with VH than in those without [34]. Additionally, significant associations between grey matter loss and cognitive functions were detected in patients with VH, specifically in measures of learning (left hippocampus, *r* = 0.88), delayed recall (left prefrontal cortex, *r* = 0.95), semantic fluency (left thalamus, *r* = 0.95), and language comprehension (left amygdala, *r* = 0.89).

In a whole brain VBM study, Ramirez-Ruiz et al. found grey matter volumetric reductions in the left lingual gyrus, and bilateral superior parietal lobe in PD patients with VH when compared with those without (*p* < 0.05 corrected at the cluster level) [30]. In this study, PD patients were matched with healthy controls. In this study, no information about the age of the included cohorts was given, and in the comparison between hallucinating and non-hallucinating patients the authors do not appear to have controlled for age in their analysis. Furthermore, patients with VH had more severe impairments in global cognitive status (as measured by the Mini-Mental State Examination, MMSE [36]), more advanced PD stage (rated with the Hoehn and Yahr scale [37]), and more severe depression (measured by the Hamilton depression rating scale [38]) [30]. These variables were included as covariates in the VBM statistical analyses [30]. Consistently with these findings, other studies reported decreased grey matter volume in PD patients with VH, compared with those without, in occipito-temporal regions, namely in the lingual and fusiform gyri, bilaterally [27,31]. Goldman et al. reported positive associations between VH severity, and grey matter volume of the left parietal lobule, and cuneus, and right lingual gyrus (*p* < 0.01 uncorrected) [27]. In addition to these findings, grey matter reductions have been reported in widespread brain areas, which were mainly located in the inferior parietal lobes [21,27], cingulate cortex [27,31], frontal [25,31], temporal [25,31], and occipital [27,31] areas bilaterally, and in the right supramarginal gyrus [21,31]. Another whole brain VBM study reported differences between PD patients with and without mild VH (presence and passage hallucinations), both decreases, mainly in the right vermis and precuneus, and increases, mainly in the cerebellum and the left inferior frontal cortex were detected in patients with mild VH [29]. These VBM studies, however, used thresholds uncorrected for multiple comparisons [21,25,27,31]. Differently from the study above, Meppelink et al. did not detect a difference between non-demented PD patients with and without VH [24], which may have been due to the more conservative threshold of *p* < 0.05 cluster-level corrected for multiple comparisons being used in this study. These apparently contrasting findings could, therefore, be a simple reflection of the application of a less rigorous statistical thresholding approach by the studies reviewed above.

In addition to the whole brain studies described above, two VBM studies investigated volumetric brain differences between hallucinating and non-hallucinating PD patients by using an ROI approach [28,32]. Specifically, Janzen et al. investigated the grey matter volume of the pedunculopontine nucleus due to its cholinergic function, thought to be involved in the development of VH, and the thalamus as one of its projection areas [28]. The authors found reduced grey matter in the left and right pedunculopontine nucleus between non-demented PD patients with and without VH, but not in the thalamus by using a threshold corrected for multiple comparisons. Hallucinating patients had significantly longer disease duration (VH: 11.5 ± 5.2 years; no VH 3.1 ± 3.6 years), were at a more severe Hoehn and Yahr stage (VH: 2.5 ± 0.3; no VH: 2.1 ± 0.5) and were taking higher levodopa equivalent doses. Furthermore, when hallucinating PD patients with and without dementia were combined and compared with non-demented patients without VH, the reduction in volume extended to the thalamus bilaterally [28].

In another voxel-based ROI study, the analyses were restricted to the hippocampus, due to its link with dementia development [32]. This region was chosen since the presence of VH is thought to be a risk factor for dementia, and therefore VH patients might exhibit the same pattern of atrophy shown by patients with dementia. The authors reported no differences in the direct comparison of non-demented PD patients with and without VH. When the two PD groups were independently compared with healthy controls, only hallucinating patients showed reduced grey matter in the anterior hippocampus bilaterally (*p* < 0.05 corrected); hippocampal volume also correlated with the learning scores achieved on a verbal memory test [32]. Overall, however, the PD patients with VH had more severe cognitive impairment, as shown by their lower MMSE scores and poorer scores on a verbal memory test [32]. Negative findings were reported by another VBM study, which found no differences in the hippocampus in any PD group, when compared with controls [35]. These contrasting findings might be due to differences in clinical and demographic variables—such as cognitive performance, disease duration, and age—and methodological differences in the analyses (i.e., standard VBM vs. DARTEL). In the latter study, when PD patients with and without VH were compared separately with healthy controls, using an ROI approach with ROI in temporal and frontal areas, reduced grey matter was found in the left superior frontal gyrus in both PD groups, while reduction in grey matter in the left frontal operculum was detected only in hallucinating patients [35]. The two PD groups, however, differed from each other in some clinical variables—including global cognitive level (measured with the MMSE), motor symptoms (evaluated with the Unified Parkinson’s Disease Rating Scale motor score subsection, UPDRS III [39]), and depression (assessed with the Beck Depression Inventory, BDI [40])—with the hallucinating group being more severe in all these measures [35]. 

Only two studies used VBM to investigate differences between hallucinating and non-hallucinating patients with DLB [26,33]. Despite the small sample size (six DLB with VH, six without VH), Sanchez-Castaneda et al. [33] found a reduction of grey matter volume in the right inferior frontal gyrus in hallucinating patients (*p* < 0.05 corrected) using a voxel-based ROI approach. In the same study, patients with PDD presented grey matter volumetric reductions in the left orbitofrontal cortex, which was no longer significant when controlling for age [33]. In the hallucinating DLB subgroup, VH severity was strongly associated with reduction in the volume of the right inferior frontal gyrus (*r* = 0.89) and left precuneus (*r* = 0.95), while no significant correlations were found in the PDD subgroup. Another VBM study in DLB identified volumetric grey matter reductions posteriorly, in the left cuneus, by using a whole brain approach [26]. These findings, however, should be taken with caution as the threshold used in this study was not corrected for multiple comparisons. Moreover, demographic and clinical comparisons between patients with and without VH were not reported, probably due to the exploratory nature of the analysis of VH (the main aim of the study was to compare DLB with Alzheimer’s disease, AD, patients and with controls) [26].

#### 3.1.2. Other Structural MRI Studies

In addition to the VBM studies described above, other studies have investigated brain morphological features of predefined ROI, especially their overall volume [18,19,20,22,24,41,42], and shape [22] in LBD. Three studies examined group differences in cortical thickness using the Freesurfer software package (and adopting thresholds corrected for multiple comparisons) [16,17,43]. Seven studies investigated volumetric [20,22,24,25,42] or cortical thickness [16,17] differences between PD patients with and without VH. Only three studies focused on the comparison between DLB and AD patients, and reported an association with VH indices [18,19,41,43].

Medial temporal lobe (MTL) structures, especially the hippocampus, were investigated by five studies [18,19,22,41,42]. Two of them focused on PD, and argued for an involvement of the hippocampus in the formation of VH mainly based on its role in memory, and evidence suggesting the presence of a high burden of Lewy body pathology in this region [22,42]. Yao et al. [22] used a multimodal MRI approach to investigate hippocampal volume, shape, mean diffusivity (MD), and functional connectivity. The authors found no differences between groups (PD with and without VH, and controls) in hippocampal volume and shape (MD and functional connectivity results are described in subsequent sections). Another MRI study reported significant volumetric reduction in hippocampal substructures, namely CA2-3 and CA4-DG, in PD patients with VH compared with those without [42]. Differences in the hippocampus as a whole were reported, however, only when hallucinating patients were compared with healthy controls. The more severe cognitive impairment in patients with VH, however, might have affected the results [42]. The value of the findings of these studies is limited by the relatively small size of the samples included in both studies [22,42]. Three MRI studies investigated the association between VH indices in DLB and MTL [18], hippocampus [19], and hippocampal substructure volumes [41]. A negative correlation was reported between severity of VHs and volumetric measures in MTL (entorhinal cortex, hippocampus, and amygdala) [18]. However, these three studies failed to report a priori hypotheses based on the involvement of MTL regions in the development of VH, probably because VH were not the primary objective of investigation.

Lee et al. [20] selected five ROI within the visual pathway to investigate differences between hallucinating and non-hallucinating PD patients in the optic chiasm area, lateral geniculate nucleus, and V1 volumes and white matter microstructure features in the optic nerve and optic radiation (the latter are described in Section 3.1.3). These regions were selected to examine the neural bases of VH in relation to their role in processing visual information. Volumetric reductions in VH patients were reported only in the lateral geniculate nucleus [20]. In addition to the whole brain analysis described in the previous section, Meppelink et al. focused on an ROI in the left fusiform gyrus, detecting no differences between PD with and without VH [24]. Finally, another structural MRI study carried out an ROI analysis by delineating the left and right substantia innominata boundaries (in addition to the VBM analysis reported above) and identified a smaller volume of this structure in hallucinating PD patients (46 PD with VH, 64 PD without VHs) [25]. Furthermore, the volume of this region correlated with scores on verbal memory, semantic fluency, and go/no-go tests [25].

Studies investigating cortical thickness did not detect any differences between PD patients with and without VH [16,17]. However, when hallucinating PD patients were compared with non-hallucinating patients at a less advanced disease stage (H&Y; PD with VH: 3.0 ± 0.5; no VH: 2.1 ± 0.4) and controls, reduced cortical thickness was reported in frontal and parietal regions [16]. In the latter study, the analysis was restricted to regions within the default mode network (DMN) [16]. On the other hand, Yao et al. [17] found no differences between PD subgroups and controls in the analysis of the whole cortical surface. Finally, Delli Pizzi et al. [43] found a significant association between the Neuropsychiatric Inventory (NPI) [44] hallucination score, and cortical thickness in right lateralised parietal regions, namely the precuneus, and superior parietal gyrus in DLB patients (*p* < 0.05 corrected). 

#### 3.1.3. Diffusion Tensor Imaging

Four DTI studies were found, two on PD [20,21], and two on DLB [45,46]. Three studies investigated predefined ROIs of grey or white matter [20,45,46], while only one used a whole brain approach, namely tract-based spatial statistics TBSS (using a threshold corrected for multiple comparisons) [21]. Most of the studies used the FMRIB software library (FLS) for DTI analysis [20,21,45]. One study did not include a sample of LBD patients without VHs, and hallucinating patients where compared to healthy controls and AD [45].

Lee et al. [20] reported disrupted white matter integrity in the right optic nerve, and in the left optic radiation by using an ROI approach. In another multimodal study (VBM analysis reviewed above), Lee et al. [21] performed voxelwise analysis of fractional anisotropy (FA) and MD by using TBSS. No differences were found between non-demented PD subgroups with and without VH, and a similar pattern of abnormalities was reported when independently compared with age-matched healthy subjects, specifically in fronto-temporo-parietal and brainstem regions [21]. In these two studies, PD with and without VH did not differ for age, disease duration, MMSE score, and motor symptoms [20,21], but in one of these studies hallucinating patients were at a more advanced disease stage than non-hallucinating patients [21]. From the articles, however, it could not be established whether some of the patients investigated in the latter two references were the same in both studies [20,21].

Although DTI is mainly used to investigate microstructural white matter abnormalities, two studies focused on grey matter [22,45]. Yao et al. [22] reported increased MD in the right hippocampus in hallucinating PD patients. Moreover, Delli Pizzi et al. [45] investigated grey matter MD differences between DLB patients with VH and healthy controls by using a tractography-based subdivision of the thalami. The authors found increased MD in thalamic sub-regions projecting to prefrontal, parieto-occipital cortex (bilaterally), amygdala (right lateralised), and motor cortex (left lateralised). Moreover, MD in the right thalamic sub-region projecting to parietal and occipital cortex was associated with severity of VHs [45]. Finally, among the studies that focused on DLB, Kantarci et al. [46] showed increased MD in the inferior longitudinal fasciculus in patients with VHs compared with those without. Demographic and clinical differences between these groups of patients were not reported, however, probably because the main purpose was to differentiate DLB and AD patients [46].

### 3.2. Functional Brain Imaging

A total of 37 studies undertook functional imaging focusing on VH in LBD, including studies using task-based and resting-state fMRI, PET, and SPECT.

Details regarding demographic, clinical, and methodological information, and imaging results related to VH are reported in Table 3 for each study.

#### 3.2.1. Task-Based fMRI

Ten studies used fMRI to identify brain activation patterns in response to simple visual stimuli [23,47,48,49,50], perception recognition tasks [51,52,53], and two single cases during VH [54,55]. The majority of them were in PD [48,49,50,51,52,53,54], while only three included patients with DLB [23,47,55]. Two studies performed different analyses on the same sample of DLB patients who had performed a visual task in the scanner [23,47]. Methodology and fMRI paradigm differed between studies, which may partly account for some inconsistencies in the findings. Other differences between studies include the threshold used to report the results, age, cognitive impairment, and duration of the disease. One study described in this section included PD patients experiencing minor VHs, including sensation of passage, presence, or misperceptions [49].

Five studies examined blood-oxygenation level-dependent (BOLD) signal in response to simple visual stimuli [23,47,48,49,50]. Specifically, they investigated the perception of moving stimuli [23,47,48], apparent motion [50], circular gratings [49], checkboards, objects [47], and stroboscopic stimulation [50]. Regions of both increased and decreased activation were found in hallucinating PD patients, compared with the non-hallucinating ones [48,49,50]. The most consistent finding was decreased activity in occipital and temporal regions [48,49,50], even though increases in occipital [49], and temporal [48] areas were also reported. One of these studies, however, had a very small sample size (three PD with VHs, three PD without VHs) [48]. In addition, reduced activity was found in the parietal and cingulate cortex [50]. On the other hand, increased activity was reported mainly in the frontal lobe [49,50]. Two studies focusing on DLB reported no correlation between BOLD signal and VH indices, but no comparison with patients without VHs was performed [23,47]. One of the latter studies reported a negative association between the NPI hallucination score and perfusion in V4, detected by using arterial spin labelling (ASL)-MRI [23].

Three studies on PD focused on perceptual recognition of complex visual stimuli [51,52,53]. In comparison with non-hallucinating patients, those with VH presented decreased activity in the right superior frontal gyrus (*p* < 0.05 cluster-level corrected) during perceptual recognition of faces [52], animals, and objects [51]. In addition to these regions, decreased activation was found in the right inferior frontal (face recognition) [52], left lingual, and bilateral fusiform gyri (animal/object recognition) [51]. In one of these studies [52], however, patients with VH had more severe cognitive impairment and behavioural performance (fMRI task) than those without, which might have partially affected the results. Shine et al. [53] identified dysfunctional connectivity in and between attention networks and the DMN during the bistable percept paradigm (BPP) in PD patients with VH [53]. During this task, patients were asked to discriminate between images containing only one perceptual interpretation (stable, e.g., a candlestick) and images containing more than one (bistable, e.g., two faces and a candlestick) [53]. In this study, PD patients were divided into two groups according to the percentage of misperceptions at the BPP. Patients performing above a previously established cut-off score [56] also presented clinically assessed VH, while those performing below did not [53].

Two fMRI studies recorded brain activity during the occurrence of visual hallucinations in single cases [54,55]. Both patients experienced complex VH, namely seeing animals [54,55] and people [54] in the MRI scanner. Howard et al. [55] scanned a DLB patient in the hallucination-free state (the patient was taking risperidone), and a second time whilst he was hallucinating (seven days after risperidone was stopped). They found decreased activation in V1 and V2 in response to photic stimulation while the patient was hallucinating compared with the hallucination-free scan [55]. On the other hand, Goetz et al. [54] performed an event-related design in order to compare hallucinating and non-hallucinating events in a patient with PD. While the patient was experiencing VH, decreased activity was reported mainly in occipito-temporal areas, but activity increased in the anterior and posterior cingulate cortex [54].

#### 3.2.2. Resting-State fMRI

Seven studies [16,17,22,57,58,59,60] performed resting-state fMRI analysis, including five statistical comparisons between patients with and without VH in PD [16,17,22,59,60]. Only two studies included a sample of hallucinating DLB patients and adopted correlational analyses [57,58]. Heterogeneity in methodology was found between studies. Specifically, independent component analysis (ICA) [16,17], ROI, and seed-based analyses [16,22,60] of functional connectivity, amplitude of low-frequency fluctuation (ALFF) [16,60], and graph analysis [58] were used. Moreover, between-study differences were detected for age, disease duration, motor symptoms, and global cognitive impairment, even though patients with and without VH were usually well matched within single studies. In addition, overall sample sizes were relatively small.

Two resting-state fMRI studies investigated differences between PD patients with and without VH in the functional connectivity of the DMN [16,17]. In both studies, the DMN was identified by performing ICA. The methodology implemented to investigate group differences in functional connectivity, however, was different. Specifically, in Franciotti et al. [16] pairwise ROI centred on the DMN were compared between groups. On the other hand, Yao et al. more broadly investigated the differences in the spatial map of the DMN [17]. Both studies reported increased functional connectivity in hallucinating patients in comparison with the non-hallucinating ones, mainly in fronto-parietal regions. Specifically, Franciotti et al. [16] detected increased connectivity between the superior frontal sulcus bilaterally with ipsilateral and contralateral parietal regions, and also between contralateral parietal regions. Yao et al. found increased activity in the right superior middle frontal lobe and bilateral precuneus and posterior cingulate gyrus within the DMN (*p* < 0.05 corrected) [17]. Both PD patients with and without VH presented a pattern of decreased functional connectivity when independently compared to healthy controls [16,17]. Furthermore, results from another resting-state fMRI study [59] were consistent with an association between VH and disrupted activity of the DMN, and other attention networks. The authors performed regression analyses to investigate the association between misperceptions at the BPP (the paradigm described above in Section 3.2.1) [56] and resting-state networks connectivity. All patients performing below a predefined BPP cut-off were also clinically classified as hallucinating. BPP error scores predicted connectivity between the ventral attention network (VAN) and the dorsal attention network (DAN), and increased connectivity within the DMN and the VAN [59].

In addition to the functional connectivity analyses reviewed above, Franciotti et al. [16] also investigated the fractional ALFF on the DMN centred ROI. Compared with non-hallucinating patients, PD with VH presented higher spectral power in fronto-parietal areas bilaterally [16]. Yao et al. [60] performed spectral analysis on the same sample in a previous study [17], reviewed above in Section 3.1.2. They found increased ALFF in VH patients in areas located in the cerebellum, temporal, and parietal lobes. Decreased ALFF was reported in occipital regions, namely the lingual gyrus, and cuneus bilaterally. These latter results were used to perform seed-based functional connectivity analysis. Compared with controls, both PD groups showed decreased functional connectivity, but in VH patients it was increased when compared with non-hallucinating patients [60]. In a multimodal MRI study, Yao et al. [22] reported both increased and decreased functional connectivity of the hippocampus in patients with VH compared with those without, using a seed-based approach. Specifically, increased connectivity was found with fronto-parietal regions, while it was decreased with occipito-temporal areas [22]. When compared to controls, however, both PD subgroups presented decreased connectivity of the hippocampus, bilaterally [22]. These latter studies performed different analyses on the same cohort of patients and this needs to be taken into account when interpreting the results [17,22,60] .

Yao et al. [22] also performed correlational analyses between cognitive measures and the regions of differential connectivity between PD groups. Specifically, the functional connectivity of the right hippocampus with right occipital, and medial temporal areas was negatively associated with visuospatial memory performance, which was in turn associated with VH severity [22]. On the other hand, other studies found no association between measures of VH and functional connectivity within regions in the DMN in PD [17], and in the temporal network in DLB [57]. Peraza et al. [58] explored functional connectivity in DLB by using a graph theory approach. They found no significant correlation between the NPI hallucination score and integrated global network measures [58]. However, they found an association with local network measures of node degree (negative for the left postcentral gyrus and positive for the putamen), and nodal betweenness centrality (negative for the right intracalcarine cortex and positive for the fusiform cortex) [58]. In another study, Peraza et al. performed secondary analyses on VHs in DLB, which showed an association between the NPI hallucination score and the left fronto-parietal, and sensory-motor networks [57]. It is not clear whether for two of the latter studies a subsample of the patients was from the same cohort of patients or not [57,58].

#### 3.2.3. Positron Emission Tomography

We identified 11 studies investigating regional cerebral glucose metabolism using PET. Among them, six focused on the differences between patients with and without VH, four in PD [61,62,63,64], and two in DLB [65,66]. Other studies examined associations between glucose metabolism and VH indices, such as severity and frequency [18,19,64,67,68]. Among the voxel-wise whole brain analyses, three out of five studies compared subgroups of patients [61,62,63], and one out of three used a correlation approach [67] and used thresholds corrected for multiple comparisons. Overall, sample sizes were relatively small.

PD patients with VH presented hypometabolism mainly in posterior regions, especially in the parietal and temporal lobes [61,62,64]. The bilateral precuneus and lingual gyrus were particularly affected [61,62]. Gasca-Salas et al. [62] also reported two smaller clusters in the right occipital lobe, while Boecker et al. [61] reported frontal hypometabolism in VH patients. In contrast, Nagano-Saito et al. [63] found frontal hypermetabolism, specifically in the left superior frontal gyrus. Discrepancies might be partially explained by differences in demographic and clinical features between studies, including age, disease duration, and global cognitive impairment. Moreover, in Boecker et al. [61], hallucinating patients were at a more advanced disease stage, and had more severe motor symptoms than non-hallucinating patients. However, UPDRS III scores were included as covariate of no interest in the statistical analysis [61]. Only two studies compared DLB subgroups and reported contrasting findings. In a whole brain analysis, Perneczky et al. [66] showed hypometabolism in right lateralised temporo-occipital and frontal regions. Although results from the latter study were not corrected for multiple comparisons, differences were only expected in regions found to be hypometabolic when compared with controls (occipital, temporo-parietal, and frontal areas) [66]. In contrast, Imamura et al. [65] reported increased regional cerebral glucose metabolic rate in temporal and parietal regions. In the latter study, however, patient groups significantly differed in MMSE scores (DLB with VH: 19.5 ± 3.9; no VH: 15.0 ± 3.0; AD: 19.7 ± 3.5) [65]. Moreover, different methods were used, namely whole brain voxel-wise comparisons [66] and ROI analyses [65]. In addition to these findings, a PET study divided DLB patients into two subgroups, based on the hypermetabolism of peri-motor areas, cerebellum, and basal ganglia. The group with more regions of hypermetabolism was associated with more frequent VHs [69].

Studies investigating correlations between glucose metabolism and VH indices mainly reported negative associations with posterior regions [18,19,64,67,68]. Specifically, occipital hypometabolism has been related to severity [67] and frequency [19,67] of VH. A negative correlation with the NPI hallucination score has also been found in parietal [68] and temporal regions [64]. Finally, Iizuka and Kameyama [18] found a negative association with the standardized uptake value ratio in the precuneus/cuneus (*r* = −0.62, *p* < 0.01), and a positive association with the cingulate island sign ratio on [18F]-Fluorodeoxyglucose (FDG)-PET (*r* = 0.44, *p* < 0.05).

#### 3.2.4. Single Photon Emission Computed Tomography

Regional cerebral blood flow in LBD patients with VHs has been investigated by nine SPECT studies, four on PD [70,71,72], five on DLB [73,74,75,76], one on PDD and DLB combined [77], and a single case on PDD [78]. Six studies examined the differences between hallucinating and non-hallucinating patients, three whole brain analyses [70,71,73] and three ROI analyses [72,74,76]. Different tracers were used to investigate cerebral blood flow using SPECT, including N-isopropyl-p-[^123^I]iodoamphetamine, [^99m^Tc]ethyl cysteinate dimer, and ^99m^Tc-HMPAO. Three studies investigated areas of association between perfusion and VH indices [73,75,77] and one study performed a SPECT scan during VH [78]. Overall, hallucinating patients showed regions of reduced brain perfusion compared with non-hallucinating patients. Only two studies reported no differences between groups [72,74].

Occipital hypoperfusion has been reported in both hallucinating PD [70] and DLB [73,76] patients, even though these studies present some limitations. For example, one only reported demographic and clinical characteristics of DLB and AD, without differentiating between patients with (*n* = 26) and without (*n* = 4) VH [76]. Moreover, Heitz et al. performed whole brain analyses without correcting the results for multiple comparisons [73]. Other ROI SPECT studies found no differences in occipital perfusion [72,74]. In another whole brain SPECT study, Oishi et al. [71] compared PD patients with (*n* = 24) and without (*n* = 41) VH. The authors found reduced cerebral blood flow in the right fusiform gyrus, which remained significant when correcting for multiple comparisons. Other temporal and parietal regions were found to be different between groups by using an uncorrected threshold [71]. In another whole brain study, O’Brien et al. [77] investigated the relationship between changes in brain perfusion and hallucinations over one year in a combined group of patients with DLB and PDD. They found a negative association with left parietal regions, namely the posterior cingulate gyrus and the precuneus (*p* < 0.05 cluster-level corrected) [77]. Another SPECT study performed factor analysis in order to investigate associations between regional cerebral blood flow and psychotic symptoms in DLB [75]. The authors showed a relationship between parietal and occipital hypoperfusion, and the sense of presence and hallucinations of people, but not of animals, insects and objects [75]. Finally, Kataoka et al. [78] described a patient with PDD having VHs during a SPECT scan, showing increased regional cerebral blood flow in the temporal lobe bilaterally, and in the left inferior frontal gyrus.

## 4. Discussion

The aim of the present review was to provide an overview of the neuroimaging findings from studies, which have investigated the neural bases of VHs in LBD by critically reviewing the current literature in the field. What emerged is that LBD patients with VH are characterized by widespread structural and functional brain abnormalities in cortical, but also subcortical regions. Given the more limited evidence in DLB and PDD than PD without dementia, it is difficult to infer disease-specific mechanisms within the LBD spectrum. A summary of the most consistent neuroimaging findings associated with VHs in LBD patients is shown in Table 1.

Overall, the most consistent finding among structural MRI studies of VH in LBD is grey matter loss in frontal areas, mainly in patients with dementia [33], and parietal and occipito-temporal regions [30] in non-demented PD patients. The presence of frontal and parietal impairment is consistent with results reported by neuropsychological studies showing more severe deficits in executive functions and visual attention in hallucinating patients [79,80,81,82]. This is in line with multifactorial models of VHs, proposing a role of visual attention deficits and disrupted engagement of attention networks in the development of VHs [5,83].

Notably, cholinergic treatment has been shown to ameliorate VHs and cognitive functioning, especially attention [84], corroborating the hypothesis of an involvement of attention dysfunction in the development of VH. Collerton et al. [5] proposed the Perception and Attention Deficit model, and suggested that VH result from the combination of impaired top-down and bottom-up processes, specifically coexisting deficits in attention and visual perception. These deficits would be supported by impaired activity in the lateral frontal cortex, and the ventral visual stream, respectively [5]. Surprisingly, given the established deficits in visual perception in LBD patients with VH [34,85,86,87], evidence of structural grey matter and brain metabolism differences in the occipital lobe is limited. Occipito-temporal and parietal grey matter loss, and reduction of cerebral blood flow were present mainly in PD [30,71]. Occipital hypoperfusion detected by SPECT has been reported [70,73,76], even though negative findings were also reported [72,74]. Discrepancies were reported in resting state FDG-PET studies. The most consistent finding is parietal and temporal glucose hypometabolism in PD with VH, even though inconsistencies were shown in frontal areas. The findings of these studies were both decreased and increased metabolism in the same regions in DLB, which might reflect demographic, clinical, and methodological differences between studies. In summary, resting state functional studies point towards hypometabolism/reduced blood flow in occipito-temporal and parietal regions in LBD patients with VH. This dysfunction in visual association regions might play a role in the genesis of VHs in LBD. This finding is further supported by the demonstration of disrupted white matter integrity in hallucinating DLB patients in the inferior longitudinal fasciculus [46], a bundle of associative fibres that connects the occipital and temporal lobes, which has been related to visual memory and perception [88].

To date, only a few studies have investigated how resting-state networks are disrupted in VH, and these studies have focused mainly on PD. Overall, increased functional connectivity in the DMN has been shown in hallucinating patients compared with those without hallucinations, while reduction in functional connectivity was a consistent finding in both PD subgroups when compared with healthy controls. Therefore, dysfunctional increased connectivity might play a significant role in the genesis of VH, especially within the DMN and fronto-parietal regions [16,17]. A speculative interpretation can be put forward, suggesting that a dysfunctional compensatory mechanism, resulting in increased functional connectivity in hallucinating patients, may foster the emergence of these symptoms. Functional abnormalities in frontal, temporo-occipital, and occipital areas have been reported by task-based fMRI studies. The direction of such alterations in the BOLD signal activity is still unclear, however, which might be due to differences in the behavioural tasks and in the stimuli used in the different studies.

Taken together, the results of imaging studies in LBD patients with VH are scarce for DLB but more frequent for PD. There is a mismatch between a more prominent involvement of primary and association visual regions in brain metabolism and blood flow studies and a more prominent involvement of more frontal regions when studying GM volume or cortical thickness. None of these findings appears to be associated with a different burden of neuropathological changes. In fact, despite the association between Lewy body pathology and VH in medial temporal lobe areas [11,12,13], substantial structural alterations in these regions have not emerged from this review. Neuropathological findings have shown a negative association between Lewy body pathology and regional brain atrophy, specifically in the frontal lobe, but conflicting evidence has been reported for the amygdala [89,90] and no associations have been found with occipital lobe dysfunction. Neither the macrostructural alterations observed with MRI nor the functional PET/SPECT findings, therefore, appear directly informative of the different underlying cellular events and neuropathology [91]. We can, therefore, speculate that VH in LBD emerge only in the presence of a double hit—i.e., concomitant alterations of large functional and structural attentional networks—of which frontal lobe atrophy may be a surrogate marker, and dysfunction of visual information processing, of which occipital-temporal and parietal hypometabolism is the functional hallmark. Large attentional networks may be impaired by diffuse cortical deposition of synuclein, and even amyloid. The cause of reduction in metabolism in posterior brain regions—i.e., which crucial cortical or subcortical projections are deafferenting the occipital cortex—remains still unexplained.

Visual hallucinations in LBD have been consistently associated with cognitive impairment. Firstly, their prevalence was found to be significantly higher in patients with dementia, and cognitive decline has been shown to be a significant predictor of VH [92,93]. In addition, there is an increased risk of developing dementia in PD in patients with early hallucinations [94]. LBD patients with VH have more severe deficits in a number of cognitive domains, especially visual perception and visual attention in both DLB [79,86,87] and PD [34,81,95,96,97], executive functioning [80,82,98,99], and long-term memory [80,81,82] in PD. Therefore, an important future development of research in this field may be the study of the association between cognitive functions and brain regions and networks specifically altered in LBD patients with VHs.

A limitation of the current literature in the field is that there have been only a few studies investigating structural brain alterations related to VH in DLB. In particular, only two studies compared patients with and without VHs directly [33,46], and no whole brain VBM analysis to date focused on VH in DLB. The neuroanatomical correlates of this symptom were assessed more extensively in PD. Further research is, therefore, needed to clarify better how these structural changes are related to cognitive functioning and connectivity between brain regions. This may be achieved by integrating studies using different imaging techniques, specifically resting-state fMRI and DTI, for the simultaneous study of functional and structural connectivity respectively. Although some resting-state fMRI studies were conducted in PD, none is available in which DLB patients with and without VHs have been directly compared. Similarly, lack of DTI studies examining white matter integrity emerged from this literature review. Future studies may benefit from the investigation of functional and structural networks associated with those cognitive functions impaired in patients with VHs. To our knowledge, only a few studies have explored the relationship between cognitive functioning and brain abnormalities in hallucinating LBD patients [22,25,32,34]. Other studies performed correlational analysis using clinical variables, especially VH severity and frequency, which have been mainly associated with parietal regions by both structural [27,33,43] and functional studies [18,57,68,77], but other correlational analyses showed no relationship with any brain region [17,19,41,46,47,58]. Furthermore, most of the studies reviewed above used the NPI questionnaire, which is not specific to capture the whole spectrum of phenomenology of VHs in this disease. Therefore, a more accurate evaluation of this symptom might be beneficial in the investigation of the neural correlates of visual hallucination. For example, the North-East Visual Hallucinations Interview (NEVHI) is a semi-structured interview, which was designed specifically to assess VH in older people with cognitive impairment [100] and this instrument might be more accurate to fine grain VH in LBD.

Finally, the present review itself presents some limitations. Even though negative results were reported by some studies, publication bias cannot be completely ruled out. In addition, we tried to reduce selection bias by undertaking an extensive literature search in two different databases with no time limit. Despite this, the possibility of having missed suitable studies cannot be fully excluded. We reviewed neuroimaging studies which had analysed VH in LBD that met the inclusion criteria. However, not all the studies had VH as their primary focus of investigation (e.g., analysis on VH in studies assessing differences between different types of dementia), and several studies failed to report clearly stated a priori hypotheses on the mechanisms underlying VH. Moreover, studies performing whole brain analyses were included even when results were not corrected for multiple comparisons, which may increase the occurrence of false positives. These factors, together with the inclusion of small sample sizes and other methodological limitations (e.g., statistical analyses not including covariates of no interest) might contribute to lowering the overall quality of the records included.

## 5. Conclusions

VHs are severe and disabling symptoms frequently observed in patients with LBD. The present review provides an up to date summary of current knowledge about the neural bases of VH in LBD. Overall, the findings suggest the involvement of structural and functional alterations in several brain areas in frontal, parietal, and occipito-temporal cortex. The mechanisms underlying VH in LBD, especially in patients with dementia, and how these differ between conditions remain still unclear, however. Future research might benefit from a combined investigation of structural and functional connectivity, as well as its association with neuropsychological measures. This might aid the understanding of the pathophysiology underlying VH in LBD and its relationship with cognitive decline. Neuroimaging techniques might help in the detection of symptom-specific biomarkers, which might be used to assess efficacy of treatments in the future, and as targets for new interventions.

## Figures and Tables

**Figure 1 brainsci-07-00084-f001:**
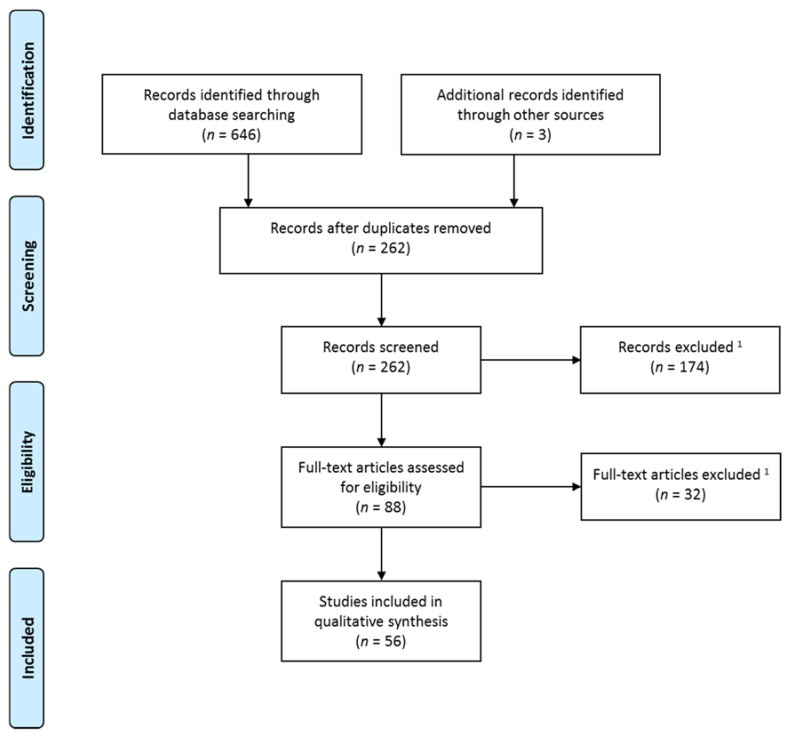
Flow chart describing the selection process of the studies included (adapted from Moher et al. [14]). ^1^ Exclusion criteria: (1) pathologies other than dementia with Lewy bodies (DLB), Parkinson’s disease (PD), or Parkinson’s disease dementia (PDD); (2) neuroimaging analysis not related to visual hallucinations (VH); (3) patients with medication-induced VH; (4) studies not using magnetic resonance imaging (MRI), functional MRI (fMRI), diffusion tensor imaging (DTI), positron emission tomography (PET), and single photon emission computed tomography (SPECT); (5) PET and SPECT studies not investigating glucose metabolism and regional cerebral blood flow; (6) MRI studies using visual rating; (7) magnetic resonance spectroscopic imaging; (8) pharmacological studies; (9) case studies (except for fMRI during VH); (10) review and theoretical articles; (11) non-English articles; and (12) non-peer reviewed articles.

**Table 1 brainsci-07-00084-t001:** Summary of the most consistent findings associated with VHs in LBD.

Brain Regions	GM Volume	Functional Connectivity	Task-Related BOLD Activation	Glucose Metabolism	Brain Perfusion
Frontal	↓	↑	↑↓	↓↑	
Parietal	↓	↑		↓↑	
Temporal				↓↑	
Occipito- temporal	↓		↓↑	↓	
Occipital			↓↑		↓

BOLD: blood-oxygenation level-dependent; GM: grey matter; LBD: Lewy body disease; VH: visual hallucinations; ↓: decrease; ↑: increase.

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
