# Peer review of "Structural and Functional Neuroimaging of Visual Hallucinations in Lewy Body Disease: A Systematic Literature Review"

_brainsci, 2017, doi:10.3390/brainsci7070084_

Round 1
Reviewer 1 Report
Line 93: How is this quality assessment defined? Is it a scale given by the authors from 0 to 10? It seemed it came from the Welton criteria. One line describing it in the section of Material and Methods would help to clarify this.
Line 544: The authors may highlight the need of using the NEVHI in clinical research for VHs.
Table A1: pro-DLB is not defined in the footnote of table A1. It stands for prodromal DLB, but a reader unfamiliar with the literature may get confused.
Author Response
Line 93: How is this quality assessment defined? Is it a scale given by the authors from 0 to 10? It seemed it came from the Welton criteria. One line describing it in the section of Material and Methods would help to clarify this.
The relevant part in text has been amended and this point should be clearer now
Line 544: The authors may highlight the need of using the NEVHI in clinical research for VHs.
This has been done and the following text added
For example, the North-East Visual Hallucinations Interview (NEVHI) is a semi-structured interview, which was designed specifically to assess VH in older people with cognitive impairment [100] and this instrument might be more accurate to fine grain VH in LBD.
Table A1: pro-DLB is not defined in the footnote of table A1. It stands for prodromal DLB, but a reader unfamiliar with the literature may get confused.
This has been amended as requested
Reviewer 2 Report
This is an impressive review of the current state of neuroimaging related to visual hallucinations (VH) in Lewy body disease (DLB). The authors have presented a thorough synthesis of the literature, and highlighted limitations and avenues for future research. I have the following comments that will hopefully be of use in further refining the manuscript:
- I think an additional table would be useful to reflect the most consistent findings across the imaging modalities. A synthesis of the overall findings is made in the discussion, however a table illustrating those would aid the reader in digesting the information and being able to easily contrast the different main findings across modalities. Perhaps something showing all the implicated brain regions and all imaging modalities, and having a tick/cross corresponding to which modalities detected abnormalities in each region.
- Regarding the disparate findings across some of the structural and functional methodologies, I wondered if the authors could have made more of an attempt to (at least speculatively) interpret these differences. For example, the neuropathological association between medial temporal lobe Lewy bodies and VH, may not be readily detectable by VBM or volumetrics, which can reflect cell loss or cell shrinkage, rather than cellular integrity. However, the presence of medial temporal lobe Lewy bodies may have significant implications on functional changes in the region. Similarly, it may be in that in the face of very minor structural alterations, there could be a robust functional compensatory response – suggesting why we might see some increased in connectivity in VH.
I appreciate that there is little work on reconciling neuroimaging findings across modalities in DLB, let alone in with reference to VH in particular. However, given the findings reported in the review it seemed like an important opportunity for the authors to offer some interpretations for why we may see these differences across imaging modalities.
- The authors may want to revise their terminology of “Quality assessment” to something like e.g., “Suitability assessment”. Only because in the quality assessment (reported in S1), the authors are not assessing the quality of those studies per se, but rather their suitability for inclusion in this systematic review.
- In the introduction the authors mention complex VH as well as presence sensations as a mild/early form of hallucination. They may want to make reference to illusions/misperceptions also, as these form a sizeable portion of the hallucinations spectrum in DLB/PD.
- Some of the paragraphs are quite long (e.g., 2nd paragraph of the discussion; paragraph beginning line 169), and it could improve the readability of the manuscript to break these up.
Author Response
This is an impressive review of the current state of neuroimaging related to visual hallucinations (VH) in Lewy body disease (DLB). The authors have presented a thorough synthesis of the literature, and highlighted limitations and avenues for future research. I have the following comments that will hopefully be of use in further refining the manuscript:
- I think an additional table would be useful to reflect the most consistent findings across the imaging modalities. A synthesis of the overall findings is made in the discussion, however a table illustrating those would aid the reader in digesting the information and being able to easily contrast the different main findings across modalities. Perhaps something showing all the implicated brain regions and all imaging modalities, and having a tick/cross corresponding to which modalities detected abnormalities in each region.
A table with the most consistent findings (table 1) was created and included in the discussion.
- Regarding the disparate findings across some of the structural and functional methodologies, I wondered if the authors could have made more of an attempt to (at least speculatively) interpret these differences. For example, the neuropathological association between medial temporal lobe Lewy bodies and VH, may not be readily detectable by VBM or volumetrics, which can reflect cell loss or cell shrinkage, rather than cellular integrity. However, the presence of medial temporal lobe Lewy bodies may have significant implications on functional changes in the region. Similarly, it may be in that in the face of very minor structural alterations, there could be a robust functional compensatory response – suggesting why we might see some increased in connectivity in VH.
This suggestion has been taken on board and the relevant part of text modified accordingly. This now reads as follows:
Occipito-temporal and parietal grey matter loss, and reduction of cerebral blood flow were present mainly in PD [30,71]. Occipital hypoperfusion detected by SPECT has been reported [70,73,76], even though negative findings were also reported [72,74]. Discrepancies were reported in resting state FDG-PET studies. The most consistent finding is parietal and temporal glucose hypometabolism in PD with VH, even though inconsistencies were shown in frontal areas. The findings of these studies were both decreased and increased metabolism in the same regions in DLB, which might reflect demographic, clinical, and methodological differences between studies. In summary, resting state functional studies point towards hypometabolism/reduced blood flow in occipito-temporal and parietal regions in LBD patients with VH. This dysfunction in visual association regions might play a role in the genesis of VH in LBD. This finding is further supported by the demonstration of disrupted white matter integrity in hallucinating DLB patients in the inferior longitudinal fasciculus [46], a bundle of associative fibres that connects the occipital and temporal lobes, and has been related to visual memory and perception [88].
To date, only a few studies have investigated how resting-state networks are disrupted in VH, and these studies have focused mainly on PD. Overall, increased functional connectivity in the DMN has been shown in hallucinating patients compared with those without hallucinations, while reduction in functional connectivity was a consistent finding in both PD subgroups when compared with healthy controls. Therefore, dysfunctional increased connectivity might play a significant role in the genesis of VH, especially within the DMN and fronto-parietal regions [16,17]. A speculative interpretation can be put forward, suggesting that a dysfunctional compensatory mechanism, resulting in increased functional connectivity in hallucinating patients, may foster the emergence of these symptoms. Functional abnormalities in frontal, temporo-occipital, and occipital areas have been reported by task-based fMRI studies. The direction of such alterations in the BOLD signal activity is still unclear, however. This might be due to differences in the behavioural tasks and in the stimuli used in the different studies.
Overall, discrepancies in findings between studies using different imaging modalities or MRI sequences might simply indicate the differential ability of each of these techniques to capture a qualitatively different level of dysfunction. Volumetric analysis does not capture alterations of neuronal function at the cellular level, but can only detect neuronal loss or shrinkage. Quantification of brain volume, therefore, would not detect any difference in regions where cellular alterations do not cause cell loss. In contrast, methods that are sensitive to functional changes are more readily able to detect alterations (whether increases or decreases in connectivity, or metabolic or blood flow variations) even in the absence of a substantial volumetric change, and even minor structural changes could generate a robust functional compensatory response. On this basis, the inconsistency of findings across imaging modalities or MRI sequences would be easy to explain.
I appreciate that there is little work on reconciling neuroimaging findings across modalities in DLB, let alone in with reference to VH in particular. However, given the findings reported in the review it seemed like an important opportunity for the authors to offer some interpretations for why we may see these differences across imaging modalities.
To address the reviewer’s request the following text has been added to the discussion
Taken together, the results of imaging studies in LBD patients with VH are scarce for DLB but more frequent for PD. There is a mismatch between a more prominent involvement of primary and association visual regions in brain metabolism and blood flow studies and a more prominent involvement of more frontal regions when studying GM volume or cortical thickness. None of these findings appears to be associated with a different burden of neuropathological changes. In fact, despite the association between Lewy body pathology and VH in medial temporal lobe areas [11-13], substantial structural alterations in these regions have not emerged from this review. Neuropathological findings have shown a negative association between Lewy body pathology and regional brain atrophy, specifically in the frontal lobe, but conflicting evidence has been reported for the amygdala [89,90] and no associations have been found with occipital lobe dysfunction. Neither the macrostructural alterations observed with MRI nor the functional PET/SPECT findings, therefore, appear directly informative of the different underlying cellular events and neuropathology [91]. We can, therefore, speculate that VH in LBD emerge only in the presence of by a double hit, i.e. concomitant alterations of large functional and structural attentional networks, of which frontal lobe atrophy may be a surrogate marker, and dysfunction of visual information processing, of which occipital-temporal and parietal hypometabolism is the functional hallmark. Large attentional networks may be impaired by diffuse cortical deposition of synuclein, and even amyloid. The cause of reduction in metabolism in posterior brain regions, i.e. which crucial cortical or subcortical projections are deafferenting the occipital cortex, remains still unexplained.
- The authors may want to revise their terminology of “Quality assessment” to something like e.g., “Suitability assessment”. Only because in the quality assessment (reported in S1), the authors are not assessing the quality of those studies per se, but rather their suitability for inclusion in this systematic review.
The suggested changes have been made in text following the reviewer’s advice.
- In the introduction the authors mention complex VH as well as presence sensations as a mild/early form of hallucination. They may want to make reference to illusions/misperceptions also, as these form a sizeable portion of the hallucinations spectrum in DLB/PD.
This has been done
- Some of the paragraphs are quite long (e.g., 2nd paragraph of the discussion; paragraph beginning line 169), and it could improve the readability of the manuscript to break these up.
The whole manuscript has been reviewed with the aim to improve readability and long paragraphs avoided wherever possible.